# Incidence and predictors of tuberculosis among HIV-infected children after initiation of antiretroviral therapy in Ethiopia: A systematic review and meta-analysis

Amare Kassaw[1]*, Worku Necho Asferie[2], Molla Azmeraw[3], Demewoz Kefale[1], Gashaw Kerebih[1], Gebrehiwot Berie Mekonnen[1], Fikadie Dagnew Baye[1], Shegaw Zeleke[4], Biruk Beletew[3], Solomon Demis Kebede[2], Tigabu Munye Aytenew[4], Lakachew Yismaw Bazezew[5], Muluken Chanie Agimas[6]

1 Department of Pediatrics and Child Health Nursing, College of Health Sciences, Debre Tabor University, Debre Tabor, Ethiopia, 2 Department of Maternal and Neonatal Health Nursing, College of Health Sciences, Debre Tabor University, Debre Tabor, Ethiopia, 3 Department of Pediatrics and Child Health Nursing, College of Health Sciences, Woldia University, Woldia, Ethiopia, 4 Department of Adult Health Nursing, College of Health Sciences, Debre Tabor University, Debre Tabor, Ethiopia, 5 Department of Neonatal Health Nursing, College of Health Sciences, Debre Tabor University, Debre Tabor, Ethiopia, 6 Department of Epidemiology and Biostatics, Institute of Public Health, College of Medicine and Health Sciences, University of Gondar, Gondar, Ethiopia

* amarekassaw2009@gmail.com

## Abstract

### Background

Globally, Tuberculosis (TB) is the main cause of morbidity and mortality among infectious disease. TB and Human Immune Virus (HIV) are the two deadly pandemics which interconnected each other tragically, and jeopardize the lives of children; particularly in Sub-Saharan Africa. Therefore, this review was aimed to determine the aggregated national pooled incidence of tuberculosis among HIV- infected children and its predictors in Ethiopia.

### Methods

An electronic search engine (HINARI, PubMed, Scopus, web of science), Google scholar and free Google databases were searched to find eligible studies. Quality of the studies was checked using the Joanna Briggs Institute (JBI) quality assessment checklists for cohort studies. Heterogeneity between studies was evaluated using Cochrane Q-test and the $I^2$ statistics.

### Result

This review revealed that the pooled national incidence of tuberculosis among children with HIV after initiation of ART was 3.63% (95% CI: 2.726–4.532) per 100-person-years observations. Being Anemic, poor and fair ART adherence, advanced WHO clinical staging, missing of cotrimoxazole and isoniazid preventing therapy, low CD4 cell count, and undernutrition were significant predictors of tuberculosis incidence.

**Data Availability Statement:** All relevant data are within the manuscript and its Supporting Information files.

**Funding:** The author(s) received no specific funding for this work.

**Competing interests:** The authors have declared that no competing interests exist.

**Abbreviations:** HIV, Humane Immune Virus; AIDS, Acquired Immune Deficiency Virus; HAART, Highly Active Antiretroviral Therapy; ART, Antiretroviral Therapy; AHR, Adjusted Hazard Ratio; CI, Confidence Interval; TB, Tuberculosis; IPT, Isoniazid preventive therapy; CPT, Cotrimoxazole preventive therapy; WHO, World Health Organization.

## Conclusion

The study result indicated that the incidence of TB among HIV- infected children is still high. Therefore, parents/guardians should strictly follow and adjust nutritional status of their children to boost immunity, prevent undernutrition and opportunistic infections. Cotrimoxazole and isoniazid preventive therapy need to continually provide for HIV- infected children for the sake of enhancing CD4/immune cells, reduce viral load, and prevent from advanced disease stages. Furthermore, clinicians and parents strictly follow ART adherence.

## Introduction

Tuberculosis(TB) is an infectious disease which is caused by mycobacterium tuberculosis [1] and it is the major cause of morbidity and mortality across the globe [2–5]. According to World Health Organization (WHO) estimation in 2022, nearly 11% (1.1million) TB case were among children age less than 15 years [6].

Human immune virus (HIV) extremely aggravates the developments of tuberculosis in children [7]; particularly in HIV prevalent settings [8]. TB and HIV are the two deadly infectious diseases which interconnected each other tragically, and jeopardize the lives of children; specially in Sub-Saharan Africa [2]. HIV has an effect on TB screening, diagnosis, treatment, susceptible to infections, worsening of the disease and increase new infection/ reactivation [9,10]. On the other hand, tuberculosis shoots up the virus replication by immune activation which leads to viral load increment and further progression of advanced stage of AIDS [11].Evidence indicated that HIV-infection increase the risk of developing active TB 25–30 times compared to people with no HIV [12,13].

Worldwide, the burden of TB-HIV co-infection is declined due to antiretroviral therapy (ART), intensive infection control strategy, cotrimoxazole and isoniazid preventive therapy (IPT) [14–16]. However, the two pandemics are still the main cause of suffering, mortality and shorten the lives of children [6,17,18]. A prospective analysis of ARROW trial study in African children revealed that TB incidence is very high in children with HIV; specifically in the first three month of treatment [16]. Studies from South Africa demonstrated that incidence of tuberculosis among HIV-infected children was higher than children without HIV [19]. A similar study from Kenya disclosed that acquiring of new TB infection is increased in HIV-infected children compared to their counterparts [20]. A recent study in Ethiopia also evidenced that TB was the most common opportunistic infection(29.8%) among children with HIV [21].

Previous studies had investigated the contributing factors of tuberculosis incidence among children with HIV. Among these; low CD4 count [22,23], advanced WHO stages [24], Anemia [25], undernutrition [26], missing of cotrimoxazole and isoniazid preventive therapy [27,28], immunosuppression [29].

Globally, there are several strategies to treat, prevent and end tuberculosis. World health organization (WHO) developed a new and holistic strategy to see tuberculosis free world by 2030 [30]. Reducing the incidence of tuberculosis per 100,000 populations is also one of the targets of sustainable development goals [31]. Ethiopia on its part also endeavor to prevent, control and by far to end TB tuberculosis and TB-HIV co-infection by planning and implementing several strategies alongside with international strategies [11,32]. Despite these international and national efforts to extirpate tuberculosis, the implementation process faced numerous challenges, particularity in TB- HIV co-infected children[2].

In Ethiopia, several primary studies were conducted on the incidence and predictors of tuberculosis among children infected with HIV. However, there is no comprehensive national

level study among HIV-infected children on antiretroviral therapy (ART). Therefore, the aim of this systematic review and meta-analysis was to determine the pooled national evidence of tuberculosis incidence among children infected with human immune virus (HIV) after initiation of ART and its predictors. The study result might help to the stakeholders by indicating the current TB occurrence and transmission to evaluate the program implementation and eradicate the pandemic.

## Methods

### Reporting and PROSPERO registration

This study was reported based on the guideline of reporting systematic review and meta-analysis (PRISMA) [33] (S1 Checklist); and prospectively registered at the Prospero with a registration number of CRD42023439555.

### Search strategy

An electronic search engine (HINARI, PubMed, CINHAL, Scopus, web of science) Google scholar, free Google databases and reference lists of eligible articles were searched to find eligible studies. The authors have built and conducted for PubMed data base using keywords and MeSH terms as the following: "Incidence"[MeSH] OR proportions OR "incidence rate" OR "incidence density" AND predictors OR associated factors OR determinants OR risk factors AND "Tuberculosis"[MeSH] OR "pulmonary tuberculosis" AND "HIV-Infections"[Mesh] OR "HIV-infection" OR "HIV-positive children" OR "HIV- infected children" AND "Child"[-MeSH] OR child* OR pediatric* OR paediatric* AND "Antiretroviral Therapy, Highly Active"[MeSH] OR "anti-retroviral agents" OR "anti-retroviral agents"[MeSH] OR "antiretroviral treatment" AND Ethiopia AND ((ffrft[Filter]) AND (humans[Filter]) AND (English[Filter]) AND (2014:2024[pdat])). This search was restricted to only human participants and studies conducted between 2014 and 2024 G.C (S1 Table).

### Eligibility criteria of studies

The search results of electronic data base were exported to Endnote X8 software. Two authors (AK and MCA) have removed unrelated studies based on their title and abstract. They also screened full text articles according to pre-determined inclusion and exclusion criteria. Any disputes between the authors about eligibility of articles were resolved through discussion and other reviewer members.

### Inclusion and exclusion criteria

This review incorporated studies conducted on HIV-infected children who initiated ART and reporting tuberculosis incidence rate and its predictors. Articles published in English language between January 01/2014 and 05/2024 in Ethiopia.

The authors excluded studies that did not report tuberculosis incidence rate; articles with the same outcome and objectives, and studies that reported multi-drug or extensively drug resistant tuberculosis as outcome were excluded.

### Outcome measurement

This systematic review and meta-analysis has aimed to determine two main outcomes. The first outcome is pooled incidence of tuberculosis among HIV- infected children age less than 15 years and the second is its predictors.

## Data extraction

The data were extracted by two independent revisers (AK and WA) from included studies; and disagreement between the authors was solved by free discussion. For each included studies; first author name, publication year, study region, study design, study setting, sample size, incidence rate, person-year observation, number of new TB case and predictors' effect size (AHR) were extracted on Microsoft excel spread sheet.

## Quality assessment

Two reviewers evaluated the quality of the studies using the Joanna Briggs Institute (JBI) quality assessment checklists for cohort studies. Their inconveniences were settled by other experts from the group. The quality assessment tool has the following criteria: appropriate statistical analysis used strategies to address incomplete follow up utilized, sufficient follow up time, measurement of the outcomes in a valid and reliable way, participants free of outcome at the beginning of the study, identifying confounding and strategies to reduce it. Based on this, questions that fulfill the above criteria labeled as 1 and 0 for question that did not fulfill the criteria. Studies were considered to be low risk/high quality when scored 50% or higher on the quality assessment tools. Whereas, studies scored less than 49% on quality assessment checklist were categorized as high risk/low quality [34]. Hence in this review all articles were scored greater than 50% (S2 Checklist).

## Statistical analysis

After the extraction of relevant data on Microsoft excel, exported to STATA version 17 statistical software for analysis. The pooled incidence of tuberculosis and its predictors among HIV-infected children was estimated using random effect model using DerSimonian-Laird model weight. The existence of heterogeneity between included studies was evaluated using Cochrane Q-test and the $I^2$ statistics. Small study effect was checked through graphical (funnel plot) and statistical (Egger's) test [35]. Sub-group analysis was performed to adjust random variation in the presence of significant heterogeneity between primary studies. A leave-one–out sensitivity analysis was done to evaluate the effect of single studies on pooled estimated result.

## Ethical consideration

Ethical clearance is not applicable for this review.

# Results

## Search results

The authors identified 964 potential articles from the combined electronic database searching and repositories. After removal of 298 duplicate studies, 666 studies were remained for the next screening. Reading the title and abstract, 476 studies were excluded, and 90 studies were assessed for eligibility. Finally, only 10 studies were included for this systematic review and meta-analysis after removal of 80 studies by full text review (Fig 1).

## Characteristics of included articles

Ten studies [36–45] with retrospective cohort study design were included in this systematic review and meta-analysis to estimate the pooled incidence and predictors of tuberculosis among HIV-infected children on ART. The search was done between January 01/2014 and 05/2024 since Ethiopia updated and implemented the guideline that states all children infected with HIV should start ART regardless of CD4 count and WHO clinical stage [46]. The review

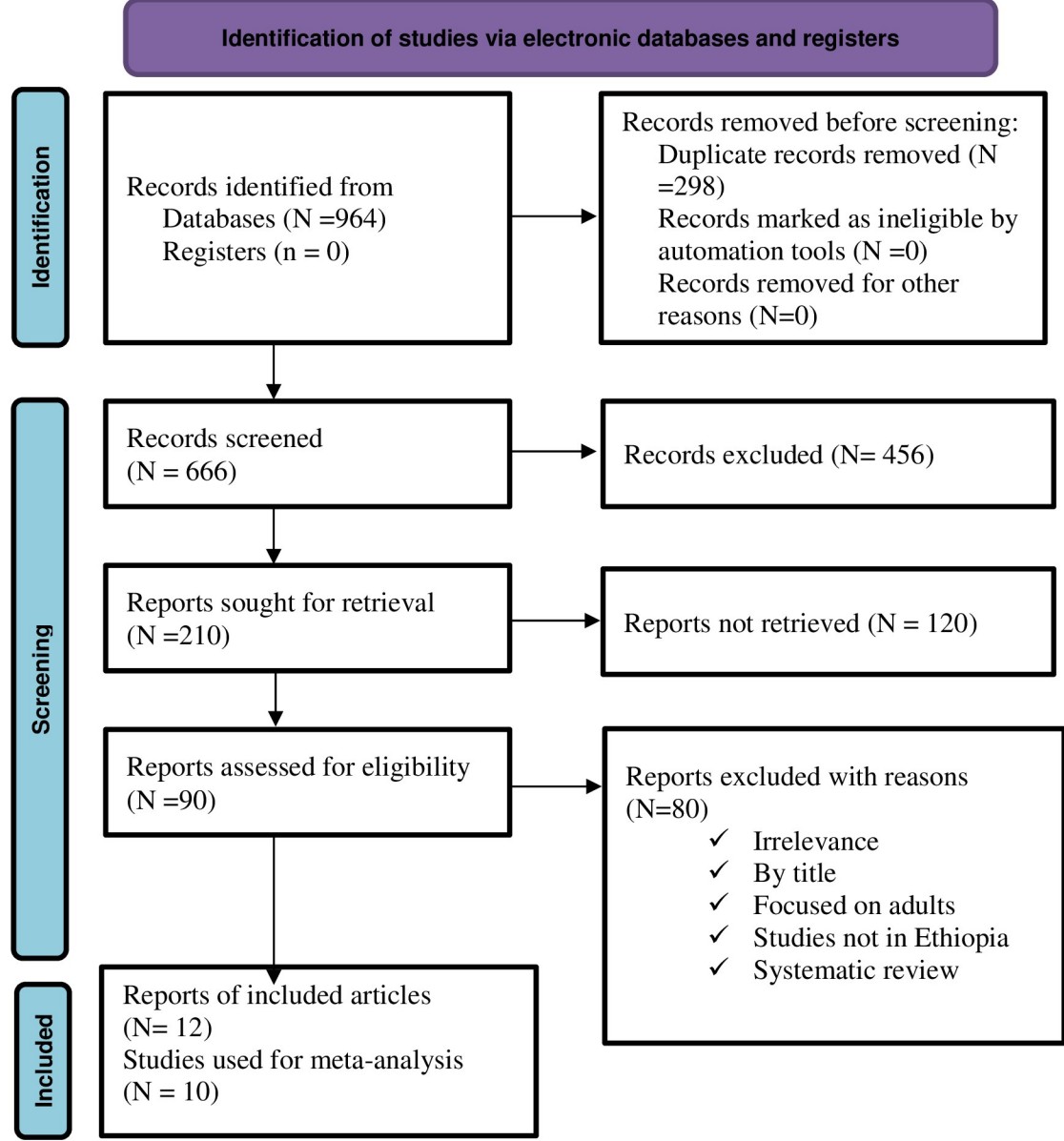

**Fig 1. PRISMA flow diagram of studies selection for systematic review and meta-analysis of the Incidence and predictors of tuberculosis among HIV-infected children in Ethiopia.**

included a total of 4881 participants, ranges from 271 a study conducted in Amhara region [40], and a sample size of 816 from study done in Adiss Ababa &Southern Nations Nationalities and Peoples Regional State (SNNPRS) [41]. The highest incidence rate of tuberculosis among HIV- infected children was observed in Benishangul Gumuz(5.9%) [36] and the lowest was in Amhara region, 2%) [43] (Table 1).

## Pooled incidence rate of tuberculosis among HIV- infected children after initiation of ART

All of the included studies were used to estimate the pooled incidence of tuberculosis among HIV-infected children on ART in Ethiopia [36–45]. Based on the random effects model, the

**Table 1. Characteristics of included studies among children on ART with outcomes.**

| Authors with publication year | Study region | Study design | Sample size | TB case | Overall PYO | IR per 100 PYO | Study quality |
|---|---|---|---|---|---|---|---|
| Alemu et al.,2016[38] | Amhara | RTCs | 645 | 79 | 1854 | 4.2 | High |
| Ayalew et al.,2015[40] | Amhara | RTCs | 271 | 52 | 1100.49 | 4.9 | High |
| Beshir et al.,2019[44] | Oromia | RTCs | 428 | 67 | 1,109.5 | 2.36 | High |
| Endalamaw et al.,2018 [45] | Amhara | RTCs | 352 | 34 | 1294.7 | 2.63 | High |
| Jerene et al.,2016[41] | A.A &SNNPRS | RTCs | 816 | 162 | 2843.53 | 2.25 | High |
| Kebede et al.,2021[36] | B. Gumuz | RTCs | 421 | 64 | 1043.1 | 5.9 | High |
| Kebede et al.,2022[39] | B. Gumuz | RTCs | 721 | 63 | 16678.07 | 5.86 | High |
| Tekese et al.,2023 [37] | SNNPRS | RTCs | 371 | 59 | 1677.5 | 3.5 | High |
| Tsegaye et al.,2023 [42] | Amhara | RTCs | 498 | 54 | - | 4.3 | High |
| Wondifraw et al.,2022 [43] | Amhara | RTCs | 358 | 79 | 2452 | 2.0 | High |

N.B, A.A = Adiss Ababa, B.Gumuz = Benishangul Gumuz, RTCs = retrospective cohort studies.

pooled incidence rate of TB was 3.63% (95% CI: 2.726–4.532) per 100-person-years observations with moderate heterogeneity ($I^2$ = 66.9, P-value <0.001) (Fig 2).

## Sub-group analyses of tuberculosis incidence among HIV-infected children on ART

From random effects model analysis, there is moderate heterogeneity ($I^2$ = 66.9%) and to handle this heterogeneity, we conducted sub-group analyses based on region, duration of follow up time and publication year. According to the region, the highest incidence was found in Benishangul Gumuz region followed by Amhara region. Studies with greater than nine years

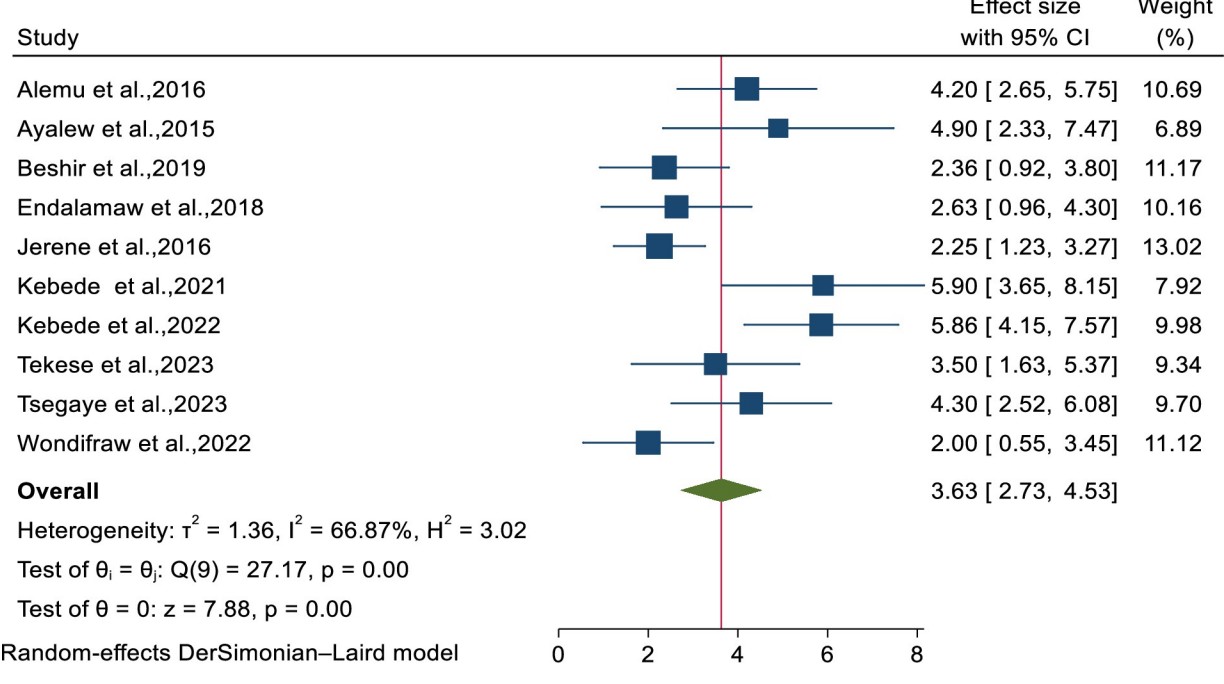

**Fig 2. Forest plots showed the incidence of tuberculosis among HIV-infected children in Ethiopia.**

**Table 2. Sub-group analyses of tuberculosis incidence among HIV-infected children on ART.**

| Categories | | Included studies | IR per 100 PYO (95% CI) | Heterogeneity (I², p-value) |
|---|---|---|---|---|
| Region | Amhara | five | 3.45 | 47.98, p<0.001 |
| | B.Gumuz | two | 5.87 | 0, p<0.001 |
| | Others | three | 2.49 | 0, p<0.001 |
| Duration of follow up time | ≥ 9 years | five | 3.36 | 56.79, p<0.001 |
| | <9 years | five | 3.89 | 75.98, p<0.001 |
| publication year | ≥2019 | Six | 3.89 | 73.43, p<0.001 |
| | <2019 | Four | 3.24 | 55.05, p<0.001 |

study follow up had relatively high incidence than less than nine years follow up. Similarly, incidence of tuberculosis was high among studies published after 2019 compared to studies published before 2019(Table 2).

## Sensitivity analysis

We have conducted leave-one-out analysis to investigate the influence of single study on the incidence of tuberculosis among HIV- infected children on ART. The analysis result revealed that there is no single study that affected the pooled estimate as overall estimate (3.63) is included within the confidence interval of all included studies (Fig 3).

## Small study effect

To investigate small study effect, we have done both Eggers' statistical test and funnel plot. Egger's statistical test indicated that there is no publication bias (p-value = 0.941). Whereas, inspection of funnel plot also evidence that there is no apparent small study effect (S1 Fig).

## Pooled estimated effects of predictors on the incidence of tuberculosis among HIV-infected children

Meta-analysis was carried out to identify cumulative estimated effects of predictors on the incidence of tuberculosis among children infected with HIV. Being Anemic, poor and fair ART

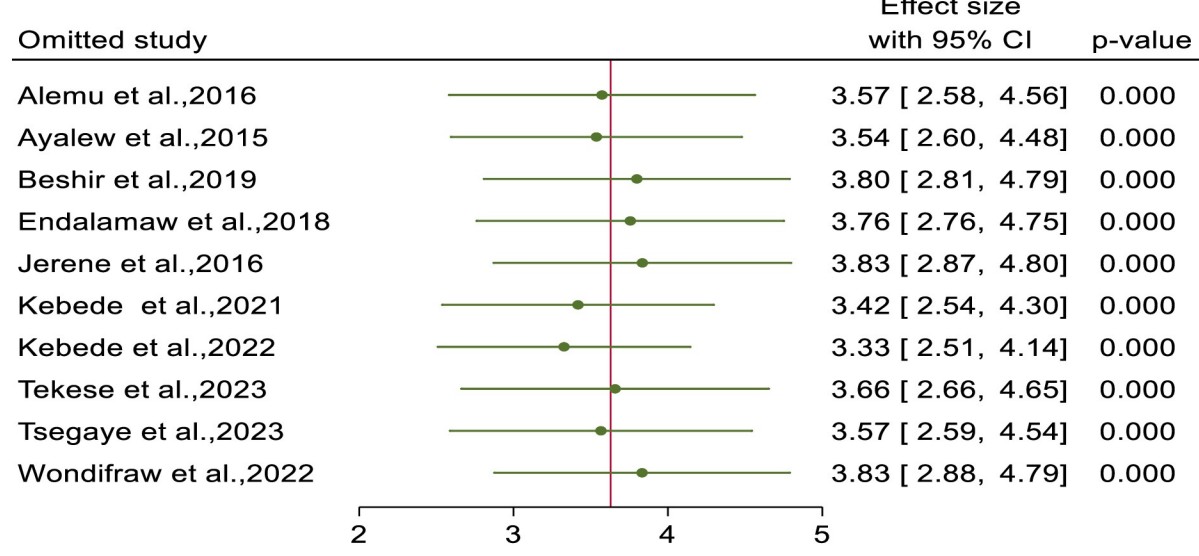

**Fig 3. Sensitivity analysis of tuberculosis incidence among HIV-infected children in Ethiopia.**

**Table 3. The pooled effects of predictors on incidence of tuberculosis among HIV-infected children in Ethiopia.**

| Variables | Categories | Observations | AHR (95% CI) | Heterogeneity(I2,P-value) | Egger's P-value |
|---|---|---|---|---|---|
| ART adherence | Fair &poor | 3 | 3.52(1.74–5.12) | 95.03, p<0.001 | 0.1079 |
| | Good | | 1 | | |
| CD4 count | Below threshold | 2 | 2.22(1.65–2.79) | 51.21, p<0.001 | 0.5623 |
| | Above threshold | | 1 | | |
| Taking CPT | No | 5 | 2.76(2.20–3.33) | 64.16, p<0.001 | 0.9531 |
| | Yes | | 1 | | |
| Anemia | Yes | 7 | 3.50(2.79–4.21) | 88.80, p<0.001 | 0.3852 |
| | No | | 1 | | |
| Taking IPT | No | 5 | 3.83(2.02–5.59) | 95.31, p<0.001 | 0.1188 |
| | Yes | | 1 | | |
| Stunting | Yes | 3 | 3.24(2.82–3.65) | 0.00, p<0.001 | 0.2039 |
| | No | | 1 | | |
| Underweight | No | 3 | 3.12(1.49–4.72) | 0.60, p<0.001 | 0.5998 |
| | Yes | | 1 | | |
| WHO clinical stages | Stage III & IV | 6 | 2.74(1.75–3.74) | 95.50, p<0.001 | 0.7033 |
| | Stage I & II | | 1 | | |

adherence, advanced WHO clinical staging, cotrimoxazole preventive therapy, isoniazid preventing therapy, low CD4 cell count, being stunting and underweight were significant predictors of tuberculosis incidence (Table 3). However, vaccination status, child developmental conditions and functional status were not associated factors of tuberculosis incidence.

## The effect of anemia on tuberculosis occurrence among children on ART

Seven studies [36–40,42,44] were used to determine the association of anemia and tuberculosis occurrence. As a result of random effects model analysis, children with hemoglobin (Hgb < 10 mg/dl) were 3.5 times (AHR = 3.50: 95% CI; 2.79–4.21) more likely to develop TB (2.79–4.21) as compared to children with hemoglobin (Hgb ≥ 10 mg/dl). There was a significant heterogeneity ($I^2$ = 88.80%, p<0.001) (Fig 4) and Egger's statistical test showed that there is no small

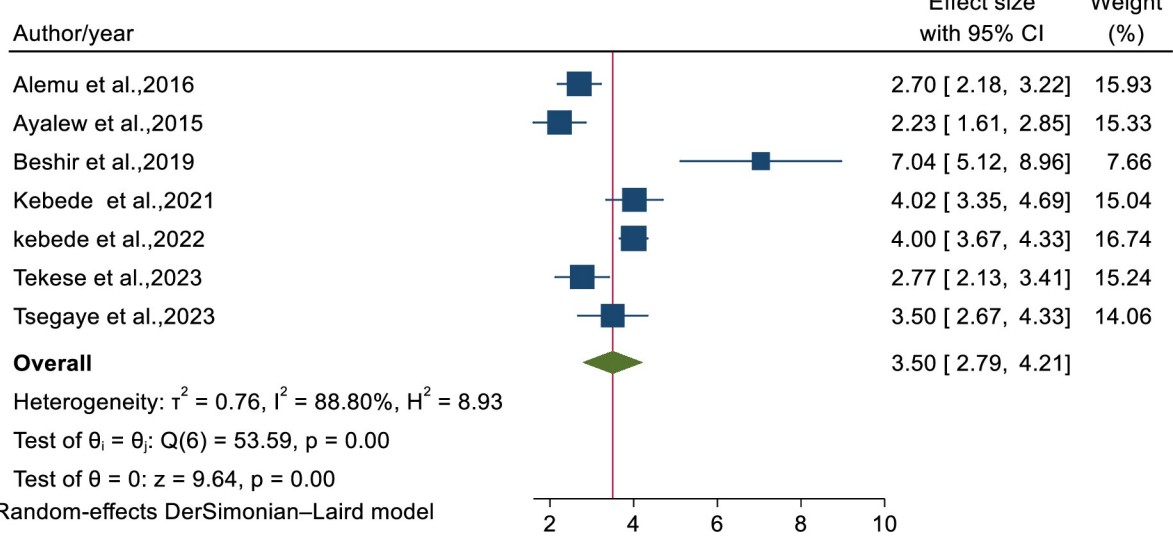

**Fig 4. Forest plot showed the association between anemia and tuberculosis incidence.**

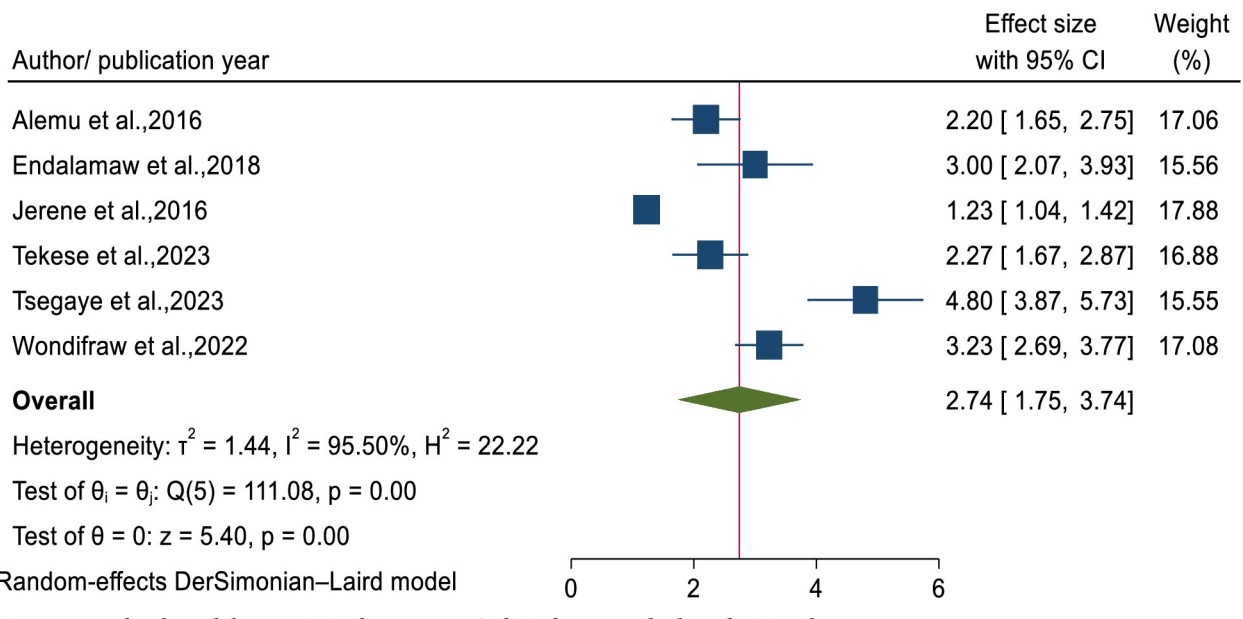

**Fig 5. Forest plot showed the association between WHO clinical stages and tuberculosis incidence.**

study effect among studies (P-value = 0.3852). From sensitivity analysis, there is no evidence of single study that excessively affect the pooled estimated effect of predictors (S2 Fig).

## The association between tuberculosis and WHO clinical stages among children on ART

Six studies [37,38,41–43,45] were examined to identify the pooled estimate of associations between tuberculosis incidence and WHO clinical stages which is categorized as stage I/II and stage III/IV for statistical analysis. A meta-analysis of random effects model indicated that HIV-infected children with WHO clinical stages (III/IV) were about 2.74 times hazard to develop tuberculosis (AHR = 2.74: 95%CI; 1.75–3.74) as compared to their counterparts. A substantial heterogeneity were observed ($I^2$ = 95.50, p<0.001) (Fig 5) with no evidence of publication bias (Egger's P-value = 0.7033). There is no a single study that excessively affect the pooled estimated effect of predictors (S3 Fig).

## Determining the association between cotrimoxazole preventive therapy on tuberculosis occurrence

Five studies [36–39,44] were included to determine the association of cotrimoxazole preventive therapy and tuberculosis incidence. The pooled result of the analysis revealed that HIV-infected children who did not CPT were 2.76 times (AHR = 2.76: 95%CI; 2.20–3.33) more likely to be infected with tuberculosis than children who provided with cotrimoxazole prophylaxis. Moderate heterogeneity was observed ($I^2$ = 64.16, p<0.001) and there is no publication bias among included articles as evidenced by (Egger's P-value = 0.5623) (Table 3).

Similarly, ART adherence status were a predictors of tuberculosis incidence among HIV-infected children. The risk of tuberculosis incidence was 3.52 times (AHR = 3.52; 95%CI: 1.74–5.12) among HIV-infected children with poor ART adherence status than good adherence; with heterogeneity ($I^2$ = 95.03%, p<0.001) and with no evidence of small study effect.

The risk of developing tuberculosis among HIV-positive children on ART was 2.22 times (AHR = 2.22; 95%CI: 1.65–2.79) whose CD4 count below the threshold (<200 cells/mm$^3$)

than children who had greater ($\geq$200 cells/mm$^3$) (I$^2$ = 51.21%, p<0.001) and with no publication bias (p = 0.5623).

The pooled effect of isoniazid preventive therapy (IPT) on the tuberculosis incidence was assessed using five studies [38,39,41,42,44]. A meta-analysis of random effects model result depicted that HIV-infected children without isoniazid preventive therapy (IPT) were about 4 times (AHR = 3.83: 95%CI; 2.02–5.59) hazard to develop tuberculosis than those who had taken IPT. Significant heterogeneity was detected (I$^2$ = 95.31, p<0.001) with no publication bias (0.1188).

From the analysis, nutritional status (stunting and underweight) were identified as a significant predictors of tuberculosis occurrence among HIV- infected children. Six studies three for stunting [36,39,43] and three for underweight [37,38,44] were used to pool the overall effect on TB incidence. HIV-infected children with stunting were 3.24 times risk to develop TB (AHR = 3.24; 95%CI: 2.82–3.65) compared to well-nourished children with no heterogeneity (I$^2$ = 0.00%, p<0.001). Likewise, being underweight has 3 times to risk of developing TB (AHR = 3.1295%CI: 1.49–4.72) with no evidence of heterogeneity (I$^2$ = 0.60, p<0.001) (Table 3).

## Discussion

This systematic review and meta-analysis was included ten primary studies conducted on incidence and predictors of tuberculosis among HIV-infected children in Ethiopia; published between 2014 and 2024. The objective of the study was to estimate the pooled incidence rate and its predictors of these individual studies.

The finding of this review showed that the pooled incidence rate of tuberculosis among children with HIV was 3.63% (95% CI: 2.726–4.532) per 100-person-years observations. The result was similar with the sub-group analysis finding of systematic review and meta- analysis conducted on both adult and children in sub-Saharan Africa [34]. However, the finding of this study was lower than studies conducted in Tanzania [29] and Nigeria [47]. The discrepancy could be socio-demographic and culture difference, introduction and implementation of ART guideline and the finding were from primary studies that might be with small sample size. On the other hand, our finding was higher than studies investigated in Thailand [48], Kenya [49], in Uganda and Zimbabwe [16] and in UK and Ireland [50]. The plausible reason for this difference might be socio-economic status (high income versus low income countries) that developed nations had better infrastructure, advanced technology and awareness for early diagnosis and managements of tuberculosis infections [51]. Other justification might be methodological and sample size variations; for instance, studies conducted in Uganda and Zimbabwe used randomized control trial.

From sub-group analysis by region indicated that the highest incidence was found in Benishangul Gumuz region followed by Amhara region. This regional variation might be due to limited number of studies, sample size and residency of study populations. Studies with >9 years study follow up period had relatively high incidence rate than <9 years follow up. This is because as the length of follow up time increased the possibility of tuberculosis/outcome occurrence will be increased. Similarly, incidence of tuberculosis was high among studies published after 2019 compared to studies published before 2019. This condition revealed that the incidence of TB is still increased even unreserved interventions had taken. The other justification might be the number of studies included before and after the year 2019.

This review disclosed that children with hemoglobin level (Hgb < 10 mg/dl) were 3.5 times more likely to develop as compared to children with hemoglobin (Hgb $\geq$ 10 mg/dl). This is consistent with the study done in Ethiopia [52], sub Saharan Africa [34,53]. The possible

justification could be; anemia in HIV- infected patients cause immune suppression and accelerate the disease progression to the advanced stage which exposed the patients to severe opportunistic infections including tuberculosis [54,55]. Other evidence from the review indicated that children with WHO clinical stages (III/IV) were about 2.74 times hazard to develop tuberculosis than stages (I/II). The finding was in line with the studies conducted in Tanzania and Nepal [56,57]. This is the fact that patients with advanced disease stages are more prone to immunosuppression which leads to variety of opportunistic infections and reactivated latent tuberculosis to active tuberculosis infection [58].

This study revealed that HIV-infected children who did not take cotrimoxazole preventive therapy (CPT) were more likely to be infected with tuberculosis than children who provided with cotrimoxazole prophylaxis which is supported by the previous studies done in Ethiopia both in adult and children [36,59,60] and other study from HIV- infected African children also confirmed that continuous provision of CPT has considerable importance in reducing TB occurrence [16]. CPT can prevent children from opportunistic infections and keep the immunity system from being eroded [61]. Evidence suggested that CPT had an action to inhibit the growth of mycobacterium tuberculosis and enhance the activity of rifampicin [62,63]. Similarly, our review also depicted that HIV-infected children who had not taken isoniazid preventive therapy (IPT) were about 4 times hazard to develop tuberculosis than those who had taken IPT. This result is consistent with the finding in Nigeria [64],Tanzania [65] and previous studies in Ethiopia [66,67]. A cluster randomized trial study conducted in Brazil showed that isoniazid preventive therapy ultimately reduce new infection of tuberculosis and related mortality of patients infected with HIV [68]. A systematic review and meta-analysis findings also supported that IPT reduce the incidence of TB by reducing reactivation of latent TB infection [69–71].

The risk of new infection of tuberculosis was 2.22 times among HIV-positive children on ART whose CD4 count below the threshold ($<200$ cells/mm$^3$) than children who had greater ($\geq 200$ cells/mm$^3$). It is in congruent with the study done in Tanzania [56], Nigeria [72], high income countries [23] and Mozambique [73]. The plausible reason is that the primary attack of HIV/AIDS infection is depletion of CD4 cells destruction and compromised the immune system that leads to various types opportunistic infections including tuberculosis [74].

The study also disclosed that HIV-infected children with poor ART adherence were more likely to be newly infected than children with good adherence status. The ultimate goal of antiretroviral therapy (ART) to reduce viral load, restoration of health, prolong survival time and scale up the living standard of HIV-infected patients [75]. However, failure to strictly adhere to ART made a conducive environment for viral replication that resulted increment of viral load and drug resistance. These condition prone the patient for impaired immunity system and thereby to develop opportunistic infections [76].

Our review indicated that HIV-infected children with undernutrion were more risk to develop tuberculosis compared to well-nourished children. The result was similar with the previous studies [16,29,76]. This is true that as immunity system weakened, the risk for acquiring of lethal opportunistic infections like tuberculosis is happened [77]. Evidence suggested that malnutrition hasten the development of tuberculosis and reactivation of latent TB infection, compromised treatment outcome and treatment failure [78].

## Strength and limitation of the study

This systematic review and meta-analysis study is the first in kind that showed the current pooled national tuberculosis incidence and its predictors among children on ART in Ethiopia. Sub-group analysis was done to explore significant heterogeneity.

Although these strengths, the study had the following limitations: the review included a small number of primary studies which minimize the strength of representativeness. Since all the included articles are retrospective studies that the data were collected from patient charts; some important predictors might be missed. Qualitative studies and articles published other than English language was excluded.

## Conclusion

The pooled national incidence of tuberculosis among HIV- infected children after initiation of ART was 3.63 per 100-person-years observations. Anemia, poor and fair ART adherence, WHO clinical staging (III/IV), children who missed take cotrimoxazole and isoniazid preventing therapy, CD4 cell count ($<200$ cells/mm$^3$), undernutrition were significant predictors of tuberculosis incidence. Parents/guardians should strictly follow and adjust nutritional status of their children to boost immunity and prevent undernutrition and opportunistic infections. Cotrimoxazole and isoniazid preventive therapy need to continually provide for HIV- infected children for the sake of enhancing CD4/immune cells, reduce viral load, and prevent from advanced disease stages. Furthermore, clinicians and parents strictly follow ART adherence.

## Supporting information

**S1 Checklist. PRISMA 2020 checklist for included articles.**
(DOCX)

**S2 Checklist. JBI critical appraisal checklist for included studies.**
(DOCX)

**S1 Fig. Funnel plot to show publication bias of the included studies.**
(TIF)

**S2 Fig. Sensitivity analysis of tuberculosis incidence for the factor anemia among HIV-infected children in Ethiopia.**
(TIF)

**S3 Fig. Sensitivity analysis of tuberculosis incidence for the factor WHO clinical stages among HIV-infected children in Ethiopia.**
(TIF)

**S1 Table. Studies search strategies and entry terms from different electronic data bases on Incidence and predictors of tuberculosis among HIV-infected children after initiation of antiretroviral therapy in Ethiopia.**
(DOCX)

**S1 Data. The data set for included studies.**
(XLSX)

## Acknowledgments

We are grateful to acknowledge Deber Tabor University for supplementary materials and support to conduct this review. We also extend our thanks to the authors of included articles in this systematic review and meta-analysis.

## Author Contributions

**Conceptualization:** Amare Kassaw, Molla Azmeraw, Muluken Chanie Agimas.

**Data curation:** Amare Kassaw, Worku Necho Asferie, Molla Azmeraw, Shegaw Zeleke, Lakachew Yismaw Bazezew, Muluken Chanie Agimas.

**Formal analysis:** Amare Kassaw, Molla Azmeraw, Gashaw Kerebih, Gebrehiwot Berie Mekonnen, Muluken Chanie Agimas.

**Funding acquisition:** Tigabu Munye Aytenew.

**Investigation:** Amare Kassaw, Fikadie Dagnew Baye, Solomon Demis Kebede.

**Methodology:** Amare Kassaw, Demewoz Kefale, Shegaw Zeleke, Muluken Chanie Agimas.

**Project administration:** Worku Necho Asferie, Gebrehiwot Berie Mekonnen, Lakachew Yismaw Bazezew.

**Resources:** Molla Azmeraw, Tigabu Munye Aytenew.

**Software:** Amare Kassaw, Demewoz Kefale, Gashaw Kerebih, Gebrehiwot Berie Mekonnen, Biruk Beletew, Solomon Demis Kebede, Tigabu Munye Aytenew, Muluken Chanie Agimas.

**Supervision:** Worku Necho Asferie.

**Validation:** Fikadie Dagnew Baye.

**Visualization:** Worku Necho Asferie, Demewoz Kefale, Gashaw Kerebih, Biruk Beletew, Lakachew Yismaw Bazezew.

**Writing – original draft:** Amare Kassaw, Molla Azmeraw, Demewoz Kefale, Gashaw Kerebih, Gebrehiwot Berie Mekonnen, Fikadie Dagnew Baye, Shegaw Zeleke, Biruk Beletew, Solomon Demis Kebede, Tigabu Munye Aytenew, Lakachew Yismaw Bazezew, Muluken Chanie Agimas.

**Writing – review & editing:** Amare Kassaw, Molla Azmeraw, Demewoz Kefale, Gashaw Kerebih, Gebrehiwot Berie Mekonnen, Fikadie Dagnew Baye, Shegaw Zeleke, Biruk Beletew, Solomon Demis Kebede, Tigabu Munye Aytenew, Lakachew Yismaw Bazezew, Muluken Chanie Agimas.

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
