## [Decision Letter · Decision Letter 0]

2 Apr 2024

PONE-D-24-05371Incidence and predictors of tuberculosis among HIV-infected children after initiation of antiretroviral therapy in Ethiopia: A systematic review and meta-analysis.PLOS ONE

Dear Dr. Kassaw,

Thank you for submitting your manuscript to PLOS ONE. After careful consideration, we feel that it has merit but does not fully meet PLOS ONE’s publication criteria as it currently stands. Therefore, we invite you to submit a revised version of the manuscript that addresses the points raised during the review process.

We look forward to receiving your revised manuscript.

Kind regards,

Zewdu Gashu Dememew, M.D

Academic Editor

Additional Editor Comments:

Dear Authors,

Congratulation to come up with this very relevant manuscript of childhood TB among HIV infected children on ART in Ethiopia.

It is well narrated, reviewed, analyzed and discussed.

Please just attend to a few comments from the one of the revue

Regards

Reviewers' comments:

Reviewer's Responses to Questions

**Comments to the Author**

1. Is the manuscript technically sound, and do the data support the conclusions?

Reviewer #1: Yes

Reviewer #2: Yes

2. Has the statistical analysis been performed appropriately and rigorously? 

Reviewer #1: Yes

Reviewer #2: Yes

3. Have the authors made all data underlying the findings in their manuscript fully available?

Reviewer #1: Yes

Reviewer #2: Yes

4. Is the manuscript presented in an intelligible fashion and written in standard English?

Reviewer #1: No

Reviewer #2: Yes

5. Review Comments to the Author

Reviewer #1: • The publication by Amare Kassaw1 et al “Incidence and predictors of tuberculosis among HIV-infected children after initiation of antiretroviral therapy in Ethiopia: A systematic review and meta-analysis’, was read and reviewed with great passion and sincerity. I note that this study was registered in PROSPERO on 06/07/2023 with registration number CRD42023439555 as required of systematic reviews.

• The critical appraisal was done using the Joanna Briggs Institute (JBI) quality assessment checklist for cohort studies as seen in the attached S2 document. This implies that high quality studies were considered since JBI is more sensitive than both CASP and ETQS. This however is dependent on the level of expertise since JBI is not user friendly for students.

• The authors do not provide a detailed explanation of eliminated studies and their disqualifying reasons. This could lead to elimination of relevant and useful studies. The authors could use reference manager to avoid deleting duplicates by one category say author’s name or year of publication.

• I note that the review considered only published articles in open access journals and went ahead to acknowledge the authors of the primary articles considered. I also note that no systematic review has ever analyzed or critiqued the used articles.

• The authors used the random effects model to do the meta-analysis which gives weight smaller but relevant studies which isn’t the case for the fixed effects model.

• The article has got a few grammatical errors (especially mixed use of upper & lower cases in the authors’ addresses) which I feel should be harmonized before its publication, either by use of an expert librarian or any other English language expert.

• I confirm that I have read this submission and can say without doubt that it is of an acceptable scientific and ethical standard for publishing in Plos one.

Reviewer #2: The Manuscript meets all the necessary guidelines to satisfy the PLOS ONE criteria for publication. The analysis is performed approriately and presented in an intelligent manner in addition to fulfilling all the other criteria. It is from this basis that I recommend publication of the manuscript.

6. PLOS authors have the option to publish the peer review history of their article (what does this mean?). If published, this will include your full peer review and any attached files.

Reviewer #1: **Yes: **IVAN AHIMBISIBWE

Reviewer #2: **Yes: **Victor Draman Afayo

---

## [Author Response · Author response to Decision Letter 0]

10 Apr 2024

Response to Reviewers and Academic editor 

Point by point responses

We would like to take this opportunity to thank the reviewers and editor for sharing their view and constructive comments. The comments were very important which further improves the quality of our manuscript. The point-by-point responses for each of the comments are provided in the following pages. Our responses are written in blue font color.

# Journal requirements

1. Please ensure that your manuscript meets PLOS ONE's style requirements, including those for file naming

Authors’ Response

We are grateful to this comment of technical relevance. Thus, we have ensured that our manuscript meets PLOS ONE's style requirements, including those for file naming 

Authors’ Response 

The authors are very grateful of these constructive comments. After read the reference lists carefully, we made some correction and addressed the incomplete one. We are sure that we have not cited retracted references, almost all references are recent. 

Additional Editor Comments: #

Congratulation to come up with this very relevant manuscript of childhood TB among HIV infected children on ART in Ethiopia. It is well narrated, reviewed, analyzed and discussed. Please just attend to a few comments from the one of the revue.

Authors’ Response 

Dear editor, really thank you very much for your constructive comments that makes us strive for further relevant works and contribution of our future career in the field. 

Response to Reviewer #1 comments 

1. The authors do not provide a detailed explanation of eliminated studies and their disqualifying reasons. This could lead to elimination of relevant and useful studies. The authors could use reference manager to avoid deleting duplicates by one category say author’s name or year of publication.

Authors’ Response

First of all, we thank you the reviewer for his constructive comments and suggestions. The authors had used EndNote X8 reference manager/citation manager to screen the included studies consciously. After export the search result to EndNote X8 software, the authors used to screen orderly based on the title, abstract, full text, setting, population etc. We worked on carefully to include all the relevant studies. Moreover the authors tried to show the detail of explanation how to screen studies in Figure 1 of page 7.

2. The article has got a few grammatical errors (especially mixed use of upper & lower cases in the authors’ addresses) which I feel should be harmonized before its publication, either by use of an expert librarian or any other English language expert.

Authors’ Response 

We accepted the comment and correct on the revised manuscript. After we have read carefully through the whole document, we properly addressed the concerned issues. 

Reviewer #2: 

The Manuscript meets all the necessary guidelines to satisfy the PLOS ONE criteria for publication. The analysis is performed appropriately and presented in an intelligent manner in addition to fulfilling all the other criteria. It is from this basis that I recommend publication of the manuscript

Authors’ Response 

Really thank you very much.

---

## [Decision Letter · Decision Letter 1]

21 Jun 2024

Incidence and predictors of tuberculosis among HIV-infected children after initiation of antiretroviral therapy in Ethiopia: A systematic review and meta-analysis.

PONE-D-24-05371R1

Dear Author,

Thank you for the correction on the given comments

We’re pleased to inform you that your manuscript has been judged scientifically suitable for publication and will be formally accepted for publication once it meets all outstanding technical requirements.

Kind regards,

Mengistu Hailemariam Zenebe, PhD

Academic Editor

PLOS ONE

Additional Editor Comments (optional):

Reviewers' comments:

Reviewer's Responses to Questions

**Comments to the Author**

1. If the authors have adequately addressed your comments raised in a previous round of review and you feel that this manuscript is now acceptable for publication, you may indicate that here to bypass the “Comments to the Author” section, enter your conflict of interest statement in the “Confidential to Editor” section, and submit your "Accept" recommendation.

Reviewer #1: All comments have been addressed

2. Is the manuscript technically sound, and do the data support the conclusions?

Reviewer #1: Yes

3. Has the statistical analysis been performed appropriately and rigorously? 

Reviewer #1: Yes

4. Have the authors made all data underlying the findings in their manuscript fully available?

Reviewer #1: Yes

5. Is the manuscript presented in an intelligible fashion and written in standard English?

Reviewer #1: Yes

6. Review Comments to the Author

Reviewer #1: All questions have been fully addressed. Thank you for all this hard work

7. PLOS authors have the option to publish the peer review history of their article (what does this mean?). If published, this will include your full peer review and any attached files.

Reviewer #1: **Yes: **IVAN AHIMBISIBWE

---

## [Editor Report · Acceptance letter]

25 Jun 2024

PONE-D-24-05371R1 

PLOS ONE

Dear Dr. Kassaw, 

I'm pleased to inform you that your manuscript has been deemed suitable for publication in PLOS ONE. Congratulations! Your manuscript is now being handed over to our production team.

Kind regards, 

on behalf of

Dr. Mengistu Hailemariam Zenebe 

Academic Editor

PLOS ONE